# Have Missing Data? Make It Miss More! Imputing Tabular Data with Masked Autoencoding

## Abstract

We present REMASKER, a novel method for imputing missing values in tabular data by extending the masked autoencoding framework. In contrast to prior work, REMASKER is both *simple* – besides the missing values (*i.e.*, naturally masked), we randomly "re-mask" another set of values, optimize the autoencoder by reconstructing this re-masked set, and apply the trained model to predict the missing values; and *effective* – with extensive evaluation on benchmark datasets, we show that REMASKER performs on par with or outperforms state-of-the-art methods in terms of both imputation fidelity and utility under various missingness settings, while its performance advantage often increases with the ratio of missing data. We further explore theoretical justification for its effectiveness, showing that REMASKER tends to learn *missingness-invariant* representations of tabular data. Our findings indicate that masked modeling represents a promising direction for further research on tabular data imputation. The code is available at: https://anonymous.4open.science/r/remasker-5E3B

## 1 Introduction

Missing values are ubiquitous in real-world tabular data due to various reasons during data collection, processing, storage, or transmission. It is often desirable to know the most likely values of missing data before performing downstream tasks (*e.g.,* classification or synthesis). To this end, intensive research has been dedicated to developing imputation methods ("imputers") that estimate missing values based on observed data (Yoon et al., 2019; Jarrett et al., 2022; Kyono et al., 2021; Stekhoven & Buhlmann, 2012; Mattei & Frellsen, 2018). Yet, imputing missing values in tabular data with high fidelity and utility remains an open problem, due to challenges including the intricate correlation across different features, the variety of missingness scenarios, and the scarce amount of available data with respect to the number of missing values.

The state-of-the-art imputers can be categorized as either *discriminative* or *generative*. The discriminative imputers, such as MissForest (Stekhoven & Buhlmann, 2012), MICE (van Buuren & Groothuis-Oudshoorn, 2011), and MIRACLE (Kyono et al., 2021), impute missing values by modeling their conditional distributions on the basis of other values. In practice, these methods are often hindered by the requirement of specifying the proper functional forms of conditional distributions and adding the set of appropriate regularizers. The generative imputers, such as GAIN (Yoon et al., 2019), MIWAE (Mattei & Frellsen, 2018), GAMIN (Yoon & Sull, 2020), and HI-VAE (Nazabal et al., 2020), estimate the joint distributions of all the features by leveraging the capacity of deep generative models and impute missing values by querying the trained models. Empirically, GAN-based methods often require a large amount of training data and suffer the difficulties of adversarial training (Goodfellow et al., 2014), while VAE-based methods often face the limitations of training through variational bounds (Zhao et al., 2022). Further, some of these methods either require complete data during training or operate on the assumptions of specific missingness patterns.

In this paper, we present REMASKER, a novel method that extends the masked autoencoding (MAE) framework (Devlin et al., 2018; He et al., 2022) to imputing missing values of tabular data. The idea of REMASKER is simple: Besides the missing values in the given dataset (*i.e.*, naturally masked), we randomly select and "re-mask" another set of values, optimize the autoencoder with the objective of reconstructing this re-masked set, and then apply the trained autoencoder to predict the missing values. Compared with the prior work, REMASKER enjoys the following desiderata: (*i*) it is instantiated

with Transformer (Vaswani et al., 2017) as its backbone, of which the self-attention mechanism is able to capture the intricate inter-feature correlation (Huang et al., 2020); (*ii*) without specific assumptions about the underlying missingness mechanisms, it is applicable to various scenarios even if complete data is unavailable; and (*iii*) as the re-masking approach naturally accounts for missing values and encourages learning high-level representations beyond low-level statistics, REMASKER works effectively even under a high ratio of missing data (*e.g.*, 0.7).

With extensive evaluation on 12 benchmark datasets under various missingness scenarios, we show that REMASKER performs on par with or outperforms 13 popular methods in terms of both imputation fidelity and utility, while its performance advantage often increases with the ratio of missing data. We further explore the theoretical explanation for its effectiveness. We find that REMASKER encourages learning *missingness-invariant* representations of tabular data, which are insensitive to missing values. Our findings indicate that, besides its success in the language and vision domains, masked modeling also represents a promising direction for future research on tabular data imputation.

## 2 RELATED WORK

Here, we survey relevant literature in three categories.

**Tabular data imputation.** The existing imputation methods can be roughly categorized as either discriminative or generative. The discriminative methods (Stekhoven & Buhlmann, 2012; van Buuren & Groothuis-Oudshoorn, 2011; Kyono et al., 2021) often specify a univariable model for each feature conditional on all others and perform cyclic regression over each target variable until convergence. Recent work has also explored adaptively selecting and configuring multiple discriminative imputers (Jarrett et al., 2022). The generative methods either implicitly train imputers as generators within the GAN framework (Yoon et al., 2019; Yoon & Sull, 2020) or explicitly train deep latent-variable models to approximate the joint distributions of all features (Mattei & Frellsen, 2018; Nazabal et al., 2020). There are also imputers based on representative-value (*e.g.*, mean, median, or frequent values) substitution (Hawthorne & Elliott, 2005), EM optimization (García-Laencina et al., 2010), matrix completion (Hastie et al., 2015), or optimal transport (Muzellec et al., 2020).

**Transformer.** Transformer has emerged as a dominating design (Vaswani et al., 2017) in the language domain, in which multi-head self-attention and MLP layers are stacked to capture both short- and long-term correlations between words. Recent work has explored the use of Transformer in the vision domain by treating each image as a grid of visual words (Dosovitskiy et al., 2020). For instance, it has been integrated into image generation models (Jiang et al., 2021; Zhang et al., 2021; Hudson & Zitnick, 2021), achieving performance comparable to CNN-based models.

**Masked autoencoding.** Autoencoding is a classical method for learning representation in a self-supervised manner (Vincent et al., 2008; Pathak et al., 2016): an encoder maps an input to its representation and a decoder reconstructs the original input. Meanwhile, masked modeling is originally proposed as a pre-training method in the language domain: by holding out a proportion of a word sequence, it trains the model to predict the masked words (Devlin et al., 2018; Radford & Narasimhan, 2018). Recent work has combined autoencoding and masked modeling in vision tasks (Dosovitskiy et al., 2020; Bao et al., 2022). Particularly, the seminal MAE (He et al., 2022) represents the state of the art in self-supervised pre-training on the ImageNet-1K benchmark.

The work is also related to that models missing data by adapting existing model architectures (Przewięźlikowski et al., 2021). To our best knowledge, this represents the first work to explore the masked autoencoding method with Transformer in the task of tabular data imputation.

## 3 REMASKER

Next, we present REMASKER, an extremely simple yet effective method for imputing missing values of tabular data. We begin by formalizing the imputation problem.

### 3.1 PROBLEM FORMALIZATION

**Incomplete data.** To model tabular data with $d$ features, we consider a $d$-dimensional random variable $\mathbf{x} \triangleq (\mathbf{x}_1, \ldots, \mathbf{x}_d) \in \mathcal{X}_1 \times \ldots \times \mathcal{X}_d$, where $\mathcal{X}_i$ is either continuous or categorical for $i \in \{1, \ldots, d\}$.

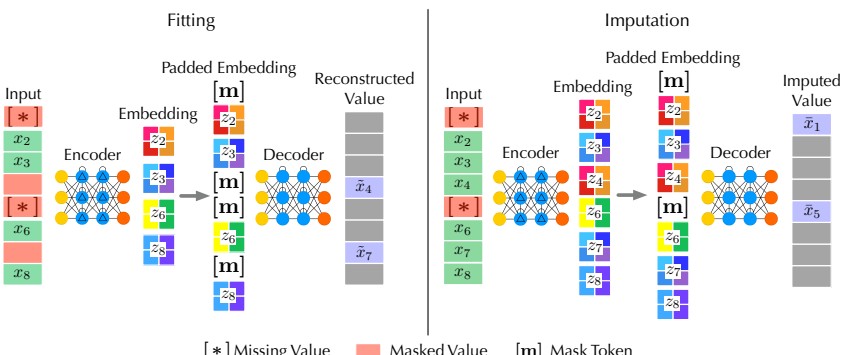

Figure 1: Overall framework of REMASKER. During the fitting stage, for each input, in addition to its missing values, another subset of values (re-masked values) is randomly selected and masked out. The encoder is applied to the remaining values to generate its embedding, which is padded with mask tokens and processed by the decoder to re-construct the re-masked values. During the imputation stage, the optimized model is applied to predict the missing values.

The observational access to $\mathbf{x}$ is mediated by an mask variable $\mathbf{m} \triangleq (\mathrm{m}_1, \ldots, \mathrm{m}_d) \in \{0, 1\}^d$, which indicates the missing values of $\mathbf{x}$, such that $\mathrm{x}_i$ is accessible only if $\mathrm{m}_i = 1$. In other words, we observe $\mathbf{x}$ in its incomplete form $\tilde{\mathbf{x}} \triangleq (\tilde{\mathrm{x}}_1, \ldots, \tilde{\mathrm{x}}_d)$ with

$$\tilde{\mathrm{x}}_i \triangleq \begin{cases} \mathrm{x}_i & \text{if } \mathrm{m}_i = 1 \\ * & \text{if } \mathrm{m}_i = 0 \end{cases} \qquad (i \in \{1, \ldots, d\}) \tag{1}$$

where $*$ denotes the unobserved value.

**Missingness mechanisms.** Missing values occur due to various reasons. To simulate different scenarios, following the prior work (Yoon et al., 2019; Jarrett et al., 2022), we consider three missingness mechanisms: MCAR ("missing completely at random") – the missingness does not depend on the data, which indicates that $\forall \mathbf{m}, \mathbf{x}, \mathbf{x}', p(\mathbf{m}|\mathbf{x}) = p(\mathbf{m}|\mathbf{x}')$; MAR ("missing at random") – the missingness depends on the observed values, which indicates that $\forall \mathbf{m}, \mathbf{x}, \mathbf{x}'$, if the observed values of $\mathbf{x}$ and $\mathbf{x}'$ are the same, then $p(\mathbf{m}|\mathbf{x}) = p(\mathbf{m}|\mathbf{x}')$; and MNAR ("missing not at random") – the missingness depends on the missing values as well, which is the case if the definitions of MCAR and MAR do not hold. In general, it is impossible to identify the missingness distribution of MNAR without domain-specific assumptions or constraints (Ma & Zhang, 2021).

**Imputation task.** In this task, we are given an incomplete dataset $\mathcal{D} \triangleq \{(\tilde{\mathbf{x}}^{(i)}, \mathbf{m}^{(i)})\}_{i=1}^n,$[1] which consists of $n$ i.i.d. realizations of $\tilde{\mathbf{x}}$ and $\mathbf{m}$. The goal is to recover the missing values of each input $\tilde{\mathbf{x}}$ by generating an imputed version $\hat{\mathbf{x}} \triangleq (\hat{\mathrm{x}}_1, \ldots, \hat{\mathrm{x}}_d)$ such that

$$\hat{\mathrm{x}}_i \triangleq \begin{cases} \tilde{\mathrm{x}}_i & \text{if } \mathrm{m}_i = 1 \\ \bar{\mathrm{x}}_i & \text{if } \mathrm{m}_i = 0 \end{cases} \qquad (i \in \{1, \ldots, d\}) \tag{2}$$

where $\bar{\mathrm{x}}_i$ is the imputed value.

### 3.2 DESIGN OF REMASKER

The REMASKER imputer extends the masked autoencoding (MAE) framework (Dosovitskiy et al., 2020; Bao et al., 2022; He et al., 2022) that reconstructs masked components based on observed components. As illustrated in Figure 1, REMASKER comprises an encoder that maps the observed values to their representations and a decoder that reconstructs the masked values from the latent representations. However, unlike conventional MAE, as the data in the imputation task is inherently incomplete (*i.e.*, naturally masked), we employ a "re-masking" approach that explicitly accounts for this incompleteness in applying masking and reconstruction. At a high level, REMASKER works in two phases: *fitting* – it optimizes the model with respect to the given dataset, and *imputation* – it applies the trained model to predict the missing values of the dataset.

**Re-masking.** In the fitting phase, for each input $\tilde{\mathbf{x}}$, in addition to its missing values, we also randomly select and mask out another subset (*e.g.*, 25%) of $\tilde{\mathbf{x}}$'s values. Formally, letting $\mathbf{m}$ be $\tilde{\mathbf{x}}$'s mask, we

---

[1]Without ambiguity, we omit the superscript $i$ in the following notations.

---

**Algorithm 1:** REMASKER

---

**Input:** $\mathcal{D} = \{(\tilde{\mathbf{x}}^{(i)}, \mathbf{m}^{(i)})\}_{i=1}^n$: incomplete dataset; remask: re-masking function; $f_\theta, d_\vartheta$: encoder and
     decoder; max_epoch: training epochs; $\ell$: reconstruction loss
**Output:** $\hat{\mathcal{D}} = \{(\hat{\mathbf{x}}^{(i)}\}_{i=1}^n$: imputed dataset

       `// fitting phase`
1 **while** *max_epoch is not reached* **do**
2    **foreach** $(\tilde{\mathbf{x}}, \mathbf{m}) \in \mathcal{D}$ **do**
3       $\tilde{\mathbf{x}}_{\mathbf{m} \wedge \overline{\mathbf{m}'}}, \tilde{\mathbf{x}}_{\mathbf{m} \wedge \mathbf{m}'} \leftarrow$ remask$(\tilde{\mathbf{x}}, \mathbf{m})$ ;              `// remasking`
4       $\mathbf{z} \leftarrow f_\theta(\tilde{\mathbf{x}}_{\mathbf{m} \wedge \mathbf{m}'})$ ;          `// encoding unmasked values`
5       pad $\mathbf{z}$ with mask tokens;
6    update $\theta, \vartheta$ by $\nabla \ell(d_\vartheta(\{\mathbf{z}\}), \{\tilde{\mathbf{x}}_{\mathbf{m} \wedge \overline{\mathbf{m}'}}\})$ ;   `// minimizing reconstruction loss`
       `// imputation phase`
7 **foreach** $(\tilde{\mathbf{x}}, \mathbf{m}) \in \mathcal{D}$ **do**
8    $\mathbf{z} \leftarrow f_\theta(\tilde{\mathbf{x}}_\mathbf{m})$ ;          `// encoding observed values`
9    pad $\mathbf{z}$ with mask tokens;
10    $\bar{\mathbf{x}}_{\overline{\mathbf{m}}} \leftarrow d_\vartheta(\mathbf{z})$ ;          `// predicting missing values`
11    $\hat{\mathbf{x}} \leftarrow \tilde{\mathbf{x}}_\mathbf{m} \cup \bar{\mathbf{x}}_{\overline{\mathbf{m}}}$;
12 **return** $\hat{\mathcal{D}} = \{\hat{\mathbf{x}}\}$;

---

define another mask vector $\mathbf{m}' \in \{0, 1\}^d$, which is randomly sampled without replacement, following a uniform distribution. Apparently, $\mathbf{m}$ and $\mathbf{m}'$ entail three subsets:

$$\mathcal{I}_{\text{mask}} = \{i | \mathbf{m}_i = 0\} \quad \mathcal{I}_{\text{remask}} = \{i | \mathbf{m}_i = 1 \wedge \mathbf{m}'_i = 0\} \quad \mathcal{I}_{\text{unmask}} = \{i | \mathbf{m}_i = 1 \wedge \mathbf{m}'_i = 1\}$$

Let $\tilde{\mathbf{x}}_{\overline{\mathbf{m}}}$, $\tilde{\mathbf{x}}_{\mathbf{m} \wedge \overline{\mathbf{m}'}}$, and $\tilde{\mathbf{x}}_{\mathbf{m} \wedge \mathbf{m}'}$ respectively be the masked, re-masked, and unmasked values. With a sufficient number of re-masked values, in addition to the missing values, we create a challenging task that encourages the model to learn missingness-invariant representations (more details in § 5). Note that in the imputation phase, we do not apply re-masking.

**Encoder.** The encoder embeds each value using an encoding function and processes the resulting embeddings through a sequence of Transformer blocks. In implementation, we apply linear encoding function to each value $x$: $\text{enc}(x) = \mathbf{w}x + \mathbf{b}$, where $\mathbf{w}$ and $\mathbf{b}$ are learnable parameters.[2] We also add positional encoding to $x$'s embedding to force the model to memorize $x$'s position in the input (*e.g.*, the $k$-th feature): $\text{pe}(k, 2i) = \sin(k/10000^{2i/d})$, where $k$ and $i$ respectively denote $x$'s position in the input and the dimension of the embedding, and $d$ is the embedding width.

Note that the encoder is only applied to the observed values: in the fitting phase, it operates on the observed values after re-masking (*i.e.*, the unmasked set $\mathcal{I}_{\text{unmask}}$); in the imputation phase, it operates on the non-missing values (*i.e.*, the union of re-masked and unmasked sets $\mathcal{I}_{\text{unmask}} \cup \mathcal{I}_{\text{remask}}$), as illustrated in Figure 1.

**Decoder.** The REMASKER decoder is instantiated as a sequence of Transformer blocks followed by an MLP layer. Different from the encoder, the decoder operates on the embeddings of both observed and masked values. Following (Devlin et al., 2018; He et al., 2022), we use a shared, learnable mask token as the initial embedding of each masked value. The decoder first adds positional encoding to the embeddings of all the values (observed and masked), processes the embeddings through a sequence of Transformer blocks, and finally applies linear projection to map the embeddings to scalar values as the predictions. Similar to (He et al., 2022), we use an asymmetric design with a deep encoder and a shallow decoder (*e.g.*, 8 blocks versus 4 blocks), which often suffices to re-construct the masked values. Conventional MAE focuses on representation learning and uses the decoder only in the training phase. In REMASKER, the decoder is required to re-construct the missing values and is thus used in both fitting and imputation phases.

**Reconstruction loss.** Recall that the REMASKER decoder predicts the value for each input feature. We define the reconstruction loss functions as the mean square error (MSE) between the reconstructed and original values on (*i*) the re-masked set $\mathcal{I}_{\text{remask}}$ and (*ii*) unmasked set $\mathcal{I}_{\text{unmask}}$. We empirically experiment with different reconstruction loss functions (*e.g.*, only the re-masked set or both re-masked and unmasked sets).

Putting everything together, Algorithm 1 sketches the implementation of REMASKER.

---

[2]We have explored other encoding functions including periodic activation function (Gorishniy et al., 2022), which observes a slight decrease (*e.g.*, $\sim 0.01$ RMSE) in imputation performance.

## 4 EVALUATION

We evaluate the empirical performance of REMASKER in various scenarios using benchmark datasets. Our experiments are designed to answer the following key questions: (*i*) *Does* REMASKER *work?* – We compare REMASKER with a variety of state-of-the-art imputers in terms of imputation quality. (*ii*) *How does it work?* – We conduct an ablation study to assess the contribution of each component of REMASKER to its performance. (*iii*) *What is the best way of using* REMASKER*?* – We explore the use of REMASKER as a standalone imputer as well as one component of an ensemble imputer to understand its best practice.

**Datasets.** For reproducibility and comparability, similar to the prior work (Yoon et al., 2019; Jarrett et al., 2022), we use 12 real-world datasets from the UCI Machine Learning repository (Dua & Graff, 2017) with their characteristics deferred to Appendix § A.1.

**Missing mechanisms.** We consider three missingness mechanisms. In MCAR, the mask vector of each input is realized following a Bernoulli random variable with a fixed mean. In MAR, with a random subset of features fixed to be observable, the remaining features are masked using a logistic model. In MNAR, the input features of MAR are further masked following a Bernoulli random variable with a fixed mean. We use the HyperImpute platform (Jarrett et al., 2022) to simulate the above missing mechanisms.

**Baselines.** We compare REMASKER with 13 state-of-the-art imputation methods: HyperImpute (Jarrett et al., 2022), a hybrid imputer that performs iterative imputation with automatic model selection; MIWAE (Mattei & Frellsen, 2018), an autoencoder model that fits missing data by optimizing a variational bound; EM (García-Laencina et al., 2010), an iterative imputer based on expectation-maximization optimization; GAIN (Yoon et al., 2019), a generative adversarial imputation network that trains the discriminator to classify the generator's output in an element-wise manner; ICE, an iterative imputer based on regularized linear regression; MICE, an ICE-like, iterative imputer based on Bayesian ridge regression; MIRACLE (Kyono et al., 2021), an iterative imputer that refines the imputation of a baseline by simultaneously modeling the missingness generating mechanism; MissForest (Stekhoven & Buhlmann, 2012), an iterative imputer based on random forests; Mean (Hawthorne & Elliott, 2005), Median, and Frequent, which impute missing values using column-wise unconditional mean, median, and the most frequent values, respectively; Sinkhorn (Muzellec et al., 2020), an imputer trained through the optimal transport metrics of Sinkhorn divergences; and SoftImpute (Hastie et al., 2015), which performs imputation through soft-thresholded singular value decomposition.

**Metrics.** For each imputation method, we evaluate the imputation fidelity and utility by comparing its imputed data with the ground-truth data. In terms of fidelity, we mainly use two metrics: root mean square error (RMSE) to measure how the individual imputed values match the ground-truth data, and the Wasserstein distance (WD) to measure how the imputed distribution matches the ground-truth distribution. In terms of utility, we use area under the receiver operating characteristic curve (AUROC) as the metric on applicable datasets (*i.e.*, ones associated with classification tasks). In the case of multi-class classification, we use the one versus rest (OvR) setting. To be fair, we use logistic regression as the predictive model across all the cases.

### 4.1 OVERALL PERFORMANCE

We evaluate REMASKER and baseline imputers on the benchmark datasets under the MAR setting with 0.3 missingness ratio, with results summarized in Figure 2. Observe that REMASKER consistently outperforms all the baselines in terms of both fidelity (measured by RMSE and WD) and utility (measured by AUROC) across all the datasets. Recall that the benchmark datasets are collected from a variety of domains with highly varying characteristics (*cf.* Table 5): the dataset size varies from 308 to 20,060, while the number of features ranges from 7 to 57. Its superior performance across all the datasets demonstrates that REMASKER effectively models the intricate correlation among different features, even if the amount of available data is scarce. The only imputer with performance close to REMASKER is HyperImpute (Jarrett et al., 2022), which is an ensemble method that integrates multiple imputation models and automatically selects the most fitting model for each column of the given dataset. This highlights that the modeling capacity of REMASKER's masked autoencoder is comparable with ensemble models. In Appendix § B.1, we conduct a more comprehensive evaluation

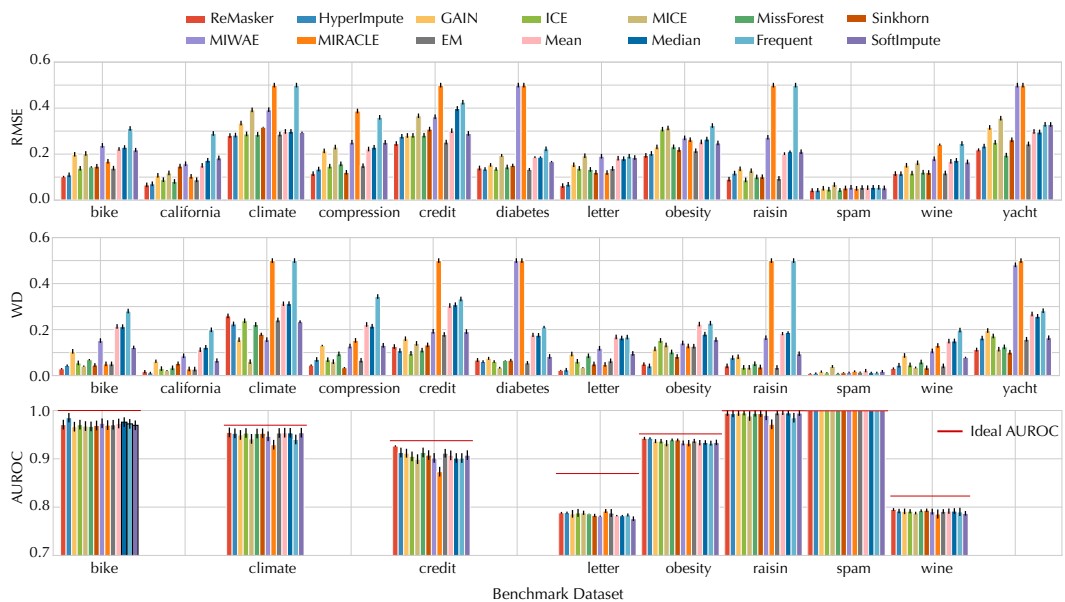

Figure 2: Overall performance of REMASKER and baseline imputers on 12 benchmark datasets under MAR with 0.3 missingness ratio. The results are shown as the mean and standard deviation of RMSE, WD, and AUROC scores (AUROC is only applicable to datasets with classification tasks). Note that REMASKER outperforms all the baseline imputers under at least one metric across all the datasets.

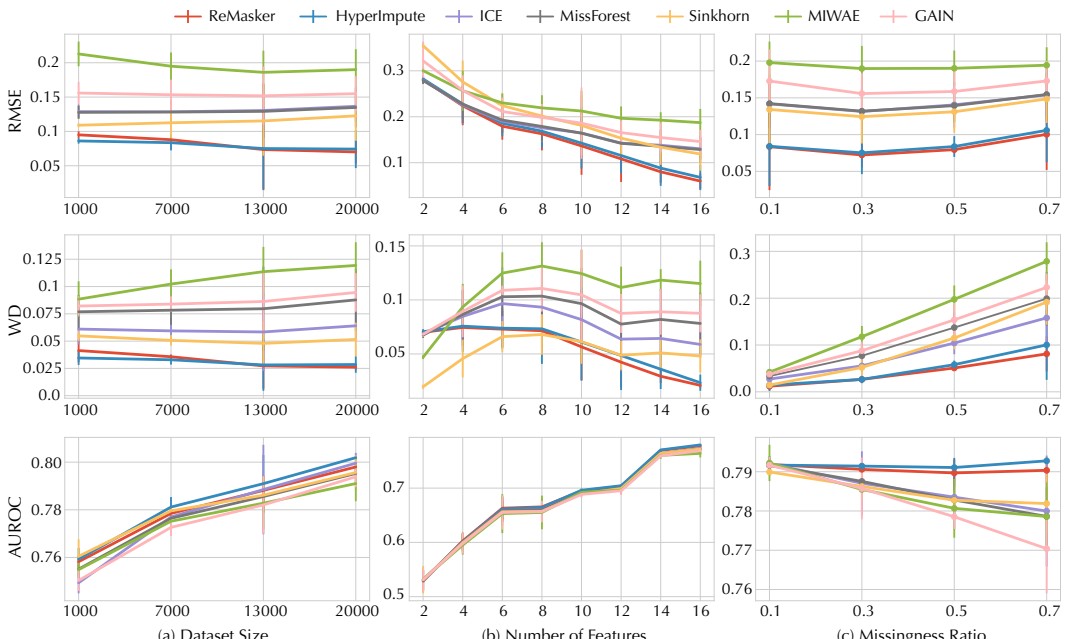

Figure 3: Sensitivity analysis of REMASKER on the `letter` dataset under the MAR setting. The results are shown in terms of RMSE, WD, and AUROC, with the scores measured with respect to (a) the dataset size, (b) the number of features, and (c) the missingness ratio. The default setting is as follows: dataset size = 20,000, number of features = 16, and missingness ratio = 0.3.

by simulating all three missingness scenarios (MCAR, MAR, and MNAR) with different missingness ratios. The results show that REMASKER consistently performs better across a range of settings.

## 4.2 SENSITIVITY ANALYSIS

To assess the factors influencing REMASKER's performance, we conduct sensitivity analysis by varying the dataset size, the number of features in the dataset, and the missingness ratio under the

MAR setting. Figure 3 shows the performance of REMASKER within these experiments against the six closest competitors (HyperImpute, ICE, MissForest, GAIN, MIWAE, and Sinkhorn) on the `letter` dataset. We have the following observations. (*a*) The performance of REMASKER improves with the size of available data, while its advantage over other imputers (with the exception of HyperImpute) grows with the dataset size. (*b*) The number of features has a significant impact on the performance of REMASKER, with its advantage over other imputers increasing steadily with the number of features. This may be explained by that REMASKER relies on learning the holistic representations of inputs, while including more features contributes to better representation learning. (*c*) REMASKER is fairly insensitive to the missingness ratio. For instance, even with 0.7 missingness ratio, it achieves RMSE below 0.1, suggesting that it effectively fits sparse datasets. In Appendix § B.2, we also conduct an evaluation on other datasets with similar observations.

## 4.3 ABLATION STUDY

We further conduct an ablation study of REMASKER to understand the contribution of different components to its performance using the `letter` dataset. The results on other datasets are deferred to Appendix § B.3.

Table 1. Ablation study of REMASKER on the `letter` dataset. The default setting is as follows: encoder depth = 8, decoder depth = 6, embedding width = 64, masking ratio = 50%, and training epochs = 600.

| depth | RMSE | WD | AUROC | width | RMSE | WD | AUROC | depth | RMSE | WD | AUROC |
|---|---|---|---|---|---|---|---|---|---|---|---|
| 2 | 0.0729 | 0.0263 | 0.7898 | 16 | 0.0902 | 0.0379 | 0.7902 | 2 | 0.0637 | 0.0239 | 0.7887 |
| 4 | 0.0636 | 0.0228 | 0.7903 | 32 | 0.0714 | 0.0289 | 0.7885 | 4 | 0.0625 | 0.0236 | 0.7877 |
| 6 | 0.0616 | 0.0219 | 0.7909 | 64 | 0.0616 | 0.0219 | 0.7909 | 6 | 0.0644 | 0.0239 | 0.7889 |
| 8 | 0.0611 | 0.0217 | 0.7892 | 128 | 0.0795 | 0.0305 | 0.7845 | 8 | 0.0616 | 0.0219 | 0.7909 |
| 10 | 0.0673 | 0.0245 | 0.7879 | 256 | 0.1040 | 0.0403 | 0.7868 | 10 | 0.0637 | 0.0227 | 0.7878 |
| | (a) Decoder depth | | | | (b) Embedding width | | | | (c) Encoder depth | | |

**Model design.** The encoder and decoder of REMASKER can be flexibly designed. Here, we study the impact of three key parameters, the encoder depth (the number of Transformer blocks in the encoder), the embedding width (the dimensionality of latent representations), and the decoder depth, with results summarized in Table 1a, Table 1b, and Table 1c, respectively. Observe that the performance of REMASKER reaches its peak with a proper model configuration (encoder depth = 8, decoder depth = 8, and embedding width = 64). This observation suggests that the model complexity needs to fit the given dataset: it needs to be sufficiently complex to effectively learn the holistic representations of inputs but not overly complex to overfit the dataset. We also compare the performance of REMASKER with different backbone models (*i.e.*, Transformer, linear, and convolutional) with the number of layers and the size of each layer fixed as the default setting. As shown in Table 2, Transformer-based REMASKER largely outperforms the other variants, which may be explained by that the self-attention mechanism of Transformer is able to effectively capture the intricate inter-feature correlation under limited data Huang et al. (2020).

Table 2. Performance of REMASKER with different backbone models.

| backbone | letter | | | california | |
|---|---|---|---|---|---|
| | RMSE | WD | AUROC | RMSE | WD |
| Transformer | 0.0611 | 0.0217 | 0.7892 | 0.0663 | 0.0172 |
| Linear | 0.1732 | 0.1604 | 0.7821 | 0.1786 | 0.1329 |
| Convolutional | 0.1694 | 0.1582 | 0.7836 | 0.1715 | 0.1286 |

Table 3. Performance of REMASKER with reconstruction loss w/ or w/o unmasked values (note: AUROC is inapplicable to the `california` dataset).

| loss | letter | | | california | |
|---|---|---|---|---|---|
| | RMSE | WD | AUROC | RMSE | WD |
| $\mathcal{I}_{\text{mask}+} \cup \mathcal{I}_{\text{unmask}}$ | 0.0616 | 0.0219 | 0.7909 | 0.0663 | 0.0172 |
| $\mathcal{I}_{\text{mask}+}$ | 0.0629 | 0.0237 | 0.7890 | 0.0840 | 0.0311 |
| $\mathcal{I}_{\text{unmask}}$ | 0.2079 | 0.1129 | 0.7901 | 0.1932 | 0.1906 |

**Reconstruction loss.** We define the reconstruction loss as the error between the reconstructed and original values on the re-masked values $\mathcal{I}_{\text{mask}+}$ and the unmasked values $\mathcal{I}_{\text{unmask}}$. We measure the performance of REMASKER under three different settings of the construction loss: (*i*) $\mathcal{I}_{\text{mask}+} \cup$

$\mathcal{I}_{\text{unmask}}$, *(ii)* $\mathcal{I}_{\text{mask}+}$ only, and *(iii)* $\mathcal{I}_{\text{unmask}}$ only on the `letter` and `california` datasets, with results shown in Table 3. Observe that using the reconstruction of unmasked values only is insufficient and yet including the reconstruction loss of unmasked values improves the performance, which is especially the case on the `california` dataset. This finding is different from the vision domain in which computing the loss on unmasked image patches reduces accuracy (He et al., 2022). We hypothesize that this difference is explained as follows. Unlike conventional MAE, due to the naturally missing values in tabular data, relying on re-masked values provides limited supervisory signals. Moreover, while images are signals with heavy spatial redundancy (*i.e.*, a missing patch can be recovered from its neighboring patches), tabular data tends to be highly semantic and information-dense. Thus, including the construction loss of unmasked values improves the model training.

## 4.4 PRACTICE OF REMASKER

Finally, we explore the optimal practice of REMASKER.

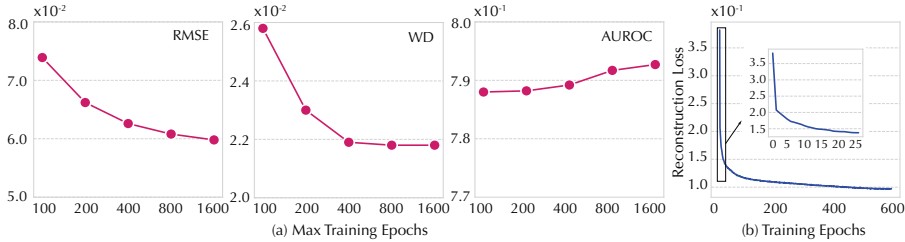

Figure 4: (a) REMASKER performance with respect to the maximum number of training epochs; (b) Convergence of REMASKER's reconstruction loss. The experiments are performed on the `letter` dataset under MAR with 0.3 missingness ratio.

**Training regime.** The ablation study above by default uses 600 training epochs. Figure 4(a) shows the impact of training epochs, in which we vary the training epochs from 100 to 1,600 and measure the performance of REMASKER on the `letter` dataset. Observe that the imputation performance improves (as RMSE and WD decrease and AUROC increases) steadily with longer training and does not fully saturate even at 1,600 epochs. However, for efficient training, it is often acceptable to terminate earlier (*e.g.*, 600 epochs) with sufficient imputation performance. To further validate the trainability of REMASKER, with the maximum number of training epochs fixed at 600 (which affects the learning rate scheduler), we measure the reconstruction loss as a function of the training epochs. As shown in Figure 4(b), the loss quickly converges to a plateau within about 100 epochs and steadily decreases after that, demonstrating the trainability of REMASKER.

**Masking ratio.** The masking ratio controls the number of re-masked values (after excluding missing values). Table 4a shows its impact on the performance of REMASKER. Observe that the optimal ratio differs across different datasets, which may be explained by the varying number of features of different datasets (16 versus 9 in `letter` and `california`). Intuitively, a larger number of features affords a higher masking ratio to balance (*i*) encouraging the model to learn missingness-invariant representations and (*ii*) having sufficient supervisory signals to facilitate the training.

Table 4. (a) REMASKER performance with respect to masking ratio; (b) REMASKER as the base imputer within HyperImpute. The results are evaluated on `letter` and `california` under MAR with 0.3 missingness ratio.

| masking | letter | | | california | |
|---|---|---|---|---|---|
| ratio | RMSE | WD | AUROC | RMSE | WD |
| 0.1 | 0.0668 | 0.0215 | 0.0789 | 0.0888 | 0.0230 |
| 0.3 | 0.0562 | 0.0207 | 0.7897 | 0.0654 | 0.0151 |
| 0.5 | 0.0554 | 0.0212 | 0.7935 | 0.0663 | 0.0172 |
| 0.7 | 0.0906 | 0.0366 | 0.7878 | 0.1320 | 0.0650 |

(a) Masking ratio.

| base imputer | letter | | | california | |
|---|---|---|---|---|---|
| | RMSE | WD | AUROC | RMSE | WD |
| default | 0.0564 | 0.0215 | 0.7899 | 0.0722 | 0.0134 |
| REMASKER | 0.0554 | 0.0212 | 0.7935 | 0.0702 | 0.0115 |

(b) REMASKER as a base imputer.

**Standalone vs. ensemble.** Besides using REMASKER as a standalone imputer, we explore its use as a base imputer within the ensemble imputation framework of HyperImpute, with results summarized in Table 4b. It is observed that compared with the default setting (with mean substitution as the base imputer), using REMASKER as the base imputer improves the imputation performance, suggesting another effective way of operating REMASKER.

## 5 DISCUSSION

The empirical evaluation above shows REMASKER's superior performance in imputing missing values of tabular data. Next, we provide theoretical justification for its effectiveness. By extending the siamese form of MAE (Kong & Zhang, 2022), we show that REMASKER encourages learning *missingness-invariant* representations of input data, which requires a holistic understanding of the data even in the presence of missing values.

Let $f_\theta(\cdot)$ and $d_\vartheta(\cdot)$ respectively be the encoder and decoder. For given input $\mathbf{x}$, mask $\mathbf{m}$, and re-mask $\mathbf{m}'$, the reconstruction loss of REMASKER training is given by (here we focus on the reconstruction of re-masked values):

$$\ell(\mathbf{x}, \mathbf{m}, \mathbf{m}') = \|d_\vartheta(f_\theta(\mathbf{x} \odot \mathbf{m} \odot \mathbf{m}')) \odot (1 - \mathbf{m}') \odot \mathbf{m} - \mathbf{x} \odot (1 - \mathbf{m}') \odot \mathbf{m}\|^2 \qquad (3)$$

where $\odot$ denotes element-wise multiplication. Let $\mathbf{m}^+ \triangleq \mathbf{m} \odot \mathbf{m}'$ and $\mathbf{m}^- \triangleq \mathbf{m} \odot (1 - \mathbf{m}')$. Eq (3) can be simplified as: $\ell(\mathbf{x}, \mathbf{m}^+, \mathbf{m}^-) = \|d_\vartheta(f_\theta(\mathbf{x} \odot \mathbf{m}^+)) \odot \mathbf{m}^- - \mathbf{x} \odot \mathbf{m}^-\|^2$. As the embedding dimensionality is typically much larger than the number of features, it is possible to make the autoencoder lossless. In other words, for a given encoder $f_\theta(\cdot)$, there exists a decoder $d_{\vartheta'}(\cdot)$, such that $d_{\vartheta'}(f_\theta(\mathbf{x} \odot \mathbf{m}^-)) \odot \mathbf{m}^- \approx \mathbf{x} \odot \mathbf{m}^-$. We can further re-write Eq (3) as:

$$\ell(\mathbf{x}, \mathbf{m}^+, \mathbf{m}^-) = \|d_\vartheta(f_\theta(\mathbf{x} \odot \mathbf{m}^+)) \odot \mathbf{m}^- - d_{\vartheta'}(f_\theta(\mathbf{x} \odot \mathbf{m}^-)) \odot \mathbf{m}^-\|^2$$
$$\text{s.t.} \quad \vartheta' = \arg\min_{\vartheta'} \mathbb{E}_{\mathbf{x}'}\|d_{\vartheta'}(f_\theta(\mathbf{x}' \odot \mathbf{m}^-)) \odot \mathbf{m}^- - \mathbf{x}' \odot \mathbf{m}^-\|^2 \qquad (4)$$

We now define a new distance metric $\Delta_{\vartheta,\vartheta'}(\mathbf{z}, \mathbf{z}') \triangleq \|(d_\vartheta(\mathbf{z}) - d_{\vartheta'}(\mathbf{z}')) \odot \mathbf{m}^-\|^2$. Then, Eq (3) is reformulated as:

$$\ell(\mathbf{x}, \mathbf{m}^+, \mathbf{m}^-) = \Delta_{\vartheta,\vartheta'}(f_\theta(\mathbf{x} \odot \mathbf{m}^+), f_\theta(\mathbf{x} \odot \mathbf{m}^-))$$
$$\text{s.t.} \quad \vartheta' = \arg\min_{\vartheta'} \mathbb{E}_{\mathbf{x}'}\|d_{\vartheta'}(f_\theta(\mathbf{x}' \odot \mathbf{m}^-)) \odot \mathbf{m}^- - \mathbf{x}' \odot \mathbf{m}^-\|^2 \qquad (5)$$

Note that optimizing Eq (5) essentially minimizes the difference between $\mathbf{x}$'s representations under $\mathbf{m}^+$ and $\mathbf{m}^-$ (with respect to the decoder). As $\mathbf{m}^+$ and $\mathbf{m}^-$ mask out different values, this formulation promotes learning representations insensitive to missing values.

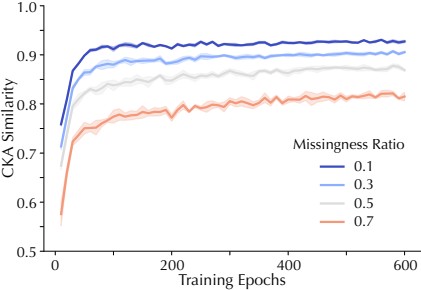

Figure 5: CKA similarity between the representations of complete and incomplete inputs (with the number of missing values controlled by the missingness ratio). The tested model is trained on `letter` under the MAR setting with 0.3 missingness ratio.

To validate the analysis above, we empirically measure the CKA similarity (Kornblith et al., 2019) between the latent representations (*i.e.*, the output of REMASKER's encoder) of complete inputs and inputs with missing values, with results shown in Figure 5. Observe that the CKA measures under different missingness ratios all steadily increase with the training length, indicating that REMASKER tends to learn missingness-invariant representations of tabular data, which may explain for its imputation effectiveness.

## 6 CONCLUSION

In this paper, we conduct a pilot study exploring the masked autoencoding approach for tabular data imputation. We present REMASKER, a novel imputation method that learns missingness-invariant representations of tabular data and effectively imputes missing values under various scenarios. With extensive evaluation on benchmark datasets, we show that REMASKER outperforms state-of-the-art methods in terms of both imputation utility and fidelity. Our findings indicate that masked tabular modeling represents a promising direction for future research on tabular data imputation.

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

## A EXPERIMENTAL DETAILS

### A.1 DATASETS

Table 5 summarizes the characteristics of the datasets used in our experiments.

### A.2 PARAMETER SETTING

The default parameter setting of REMASKER is listed in Table 6.

Table 5. Characteristics of the datasets used in the experiments.

| Dataset | Dataset Size | Number of Features | Experiment Name |
|---|---|---|---|
| California Housing | 20,640 | 9 | california |
| Climate Model Simulation Crashes | 540 | 18 | climate |
| Concrete Compressive Strength | 1,030 | 9 | compression |
| Diabetes | 442 | 10 | diabetes |
| Estimation of Obesity Levels | 2,111 | 17 | obesity |
| Credit Approval | 690 | 15 | credit |
| Wine Quality | 1,599 | 12 | wine |
| Raisin | 900 | 8 | raisin |
| Spambase | 4,601 | 57 | spam |
| Bike Sharing Demand | 8,760 | 14 | bike |
| Letter Recognition | 20,000 | 16 | letter |
| Yacht Hydrodynamics | 308 | 7 | yacht |

Table 6. Default parameter setting of REMASKER.

| model | parameter | setting |
|---|---|---|
| global | optimizer | Adam |
| | initial learning rate | 1e-3 |
| | LR scheduler | cosine annealing |
| | gradient clipping threshold | 5.0 |
| | training epochs | 600 |
| | batch size | 64 |
| | masking ratio | 0.5 |
| encoder | Transformer block | 8 |
| | embedding width | 64 |
| | number of heads | 4 |
| decoder | Transformer block | 4 |
| | embedding width | 64 |
| | number of heads | 4 |

# B  ADDITIONAL RESULTS

## B.1  OVERALL PERFORMANCE

Figure 6, 7, and 8 respectively show the imputation performance of REMASKER and 8 baselines on 12 benchmark datasets under the MAR, MCAR, and MNAR scenarios with the missingness ratio varying from 0.1 to 0.7. Observed that REMASKER performs on par with or outperforms almost all the baselines across a wide range of settings. Note that the MIRACLE imputer does not work on the `Compression` dataset and the `Raisin` dataset under some settings, of which the results are not reported. Given that both `Compression` and `Raisin` are relatively small datasets, one possible explanation is that MIRACLE requires a sufficient amount of data to train the model.

Why does REMASKER generalize across the settings of MAR, MCAR, and MNAR? One possible explanation is as follows. Recall that in MCAR, the mask vector of each input is realized following a Bernoulli random variable with a fixed mean; in MAR, with a random subset of features fixed to be observable, the remaining features are masked using a logistic model; in MNAR, the input features of MAR are further masked following a Bernoulli random variable with a fixed mean. Regardless of the missingness mechanism, it is rare that the values of one feature $x$ are missing across all the records. Thus, by its design, REMASKER is able to learn to re-construct feature $x_i$ conditional on other features $x_{\bar{i}} = (x_1, \ldots, x_{i-1}, x_{i+1}, \ldots, x_d)$. Yet, as reflected in the imputation results, the learning to re-construct performs better under MCAR, in which the missing values are evenly distributed across different features, than MAR or MNAR, in which the missing values are not evenly distributed.

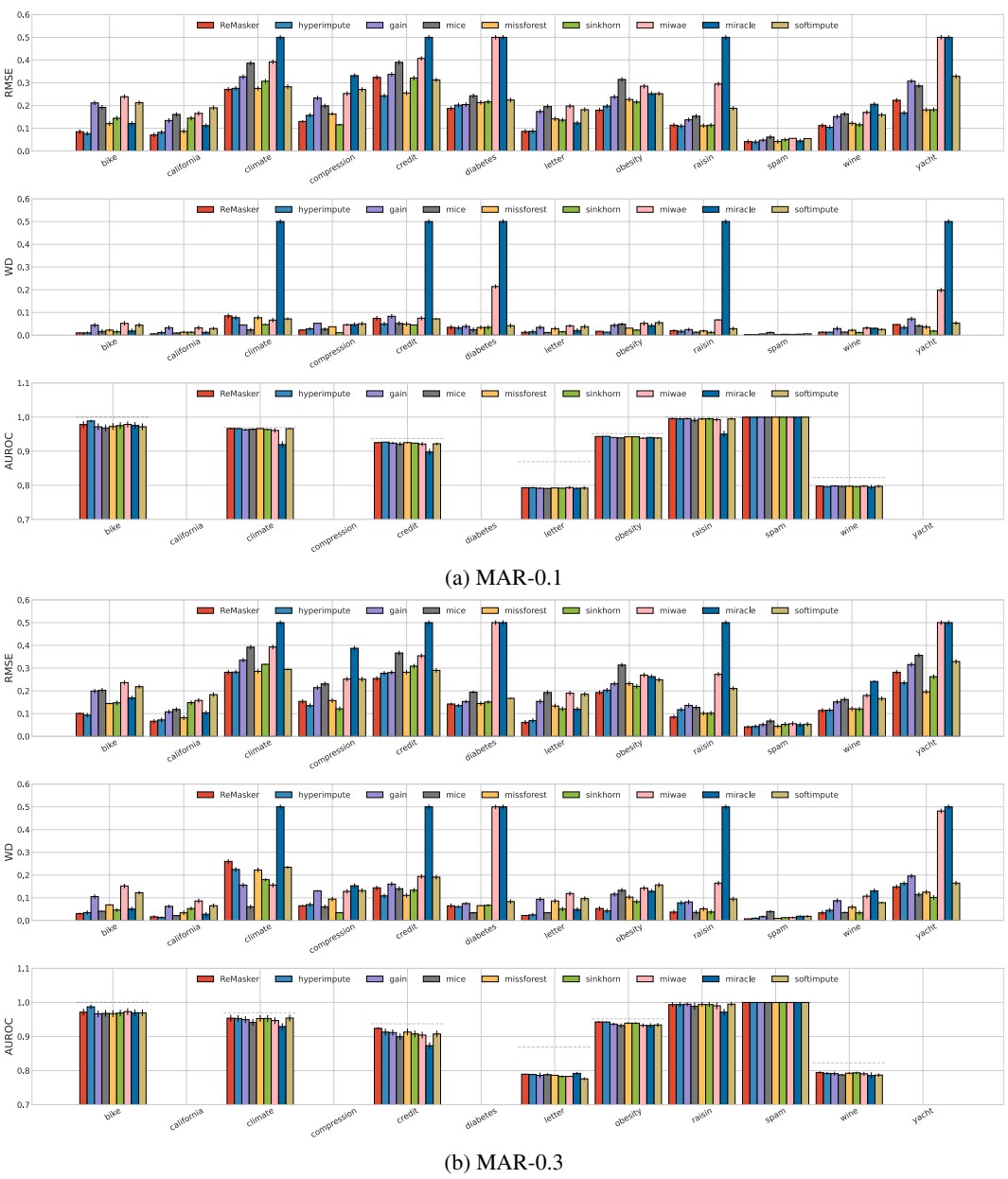

Figure 6a: Overall performance of REMASKER and 8 baselines on 12 benchmark datasets under MAR scenario with 0.1 and 0.3 missingness ratio. The results are shown as the mean and standard deviation of RMSE, WD, and AUROC scores (AUROC is only applicable to datasets with classification tasks).

## B.2 SENSITIVITY ANALYSIS

Figure 9 shows the sensitivity analysis of REMASKER and other 6 baselines on the california dataset under the MAR, MCAR, and MNAR settings. The observed trends are generally similar to that in Figure 3, which further demonstrates the observations we made in § 4 about how different factors may impact REMASKER's imputation performance.

## B.3 ABLATION STUDY

The ablation study of REMASKER on the california dataset is shown in Table 7. Observed that the performance of REMASKER reaches its peak with encoder depth = 6, decoder depth = 4, and embedding width = 32.

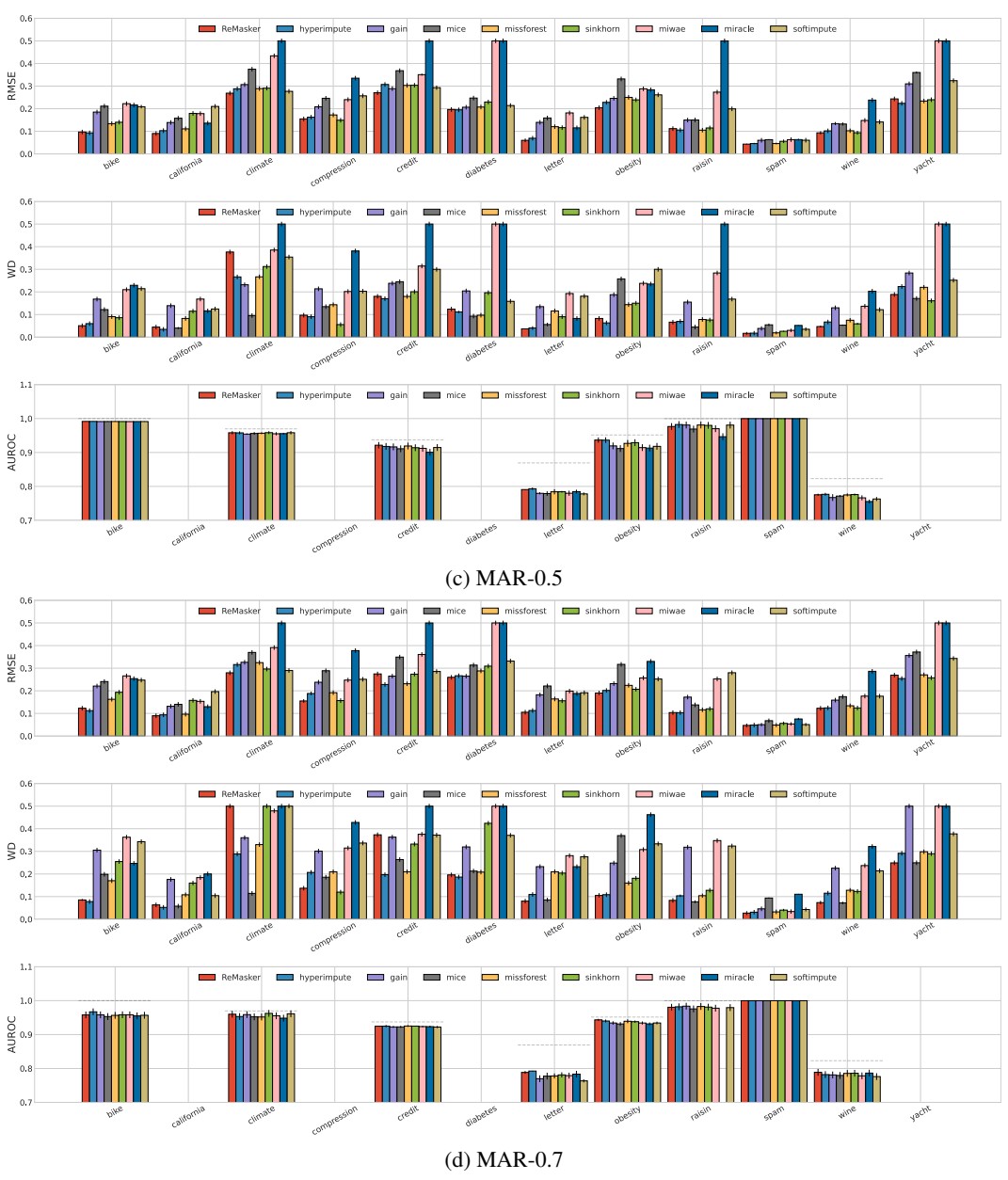

(c) MAR-0.5

(d) MAR-0.7

Figure 6b: Overall performance of REMASKER and 8 baselines on 12 benchmark datasets under MAR scenario with 0.5 and 0.7 missingness ratio. The results are shown as the mean and standard deviation of RMSE, WD, and AUROC scores (AUROC is only applicable to datasets with classification tasks).

## B.4 TRAINING REGIME

Figure 10 shows the imputation performance of REMASKER on the `california` dataset when the training length varies from 100 to 1,600 epochs. Figure 11 plots the convergence of reconstruction loss in REMASKER, showing a trend similar to Figure 4(b).

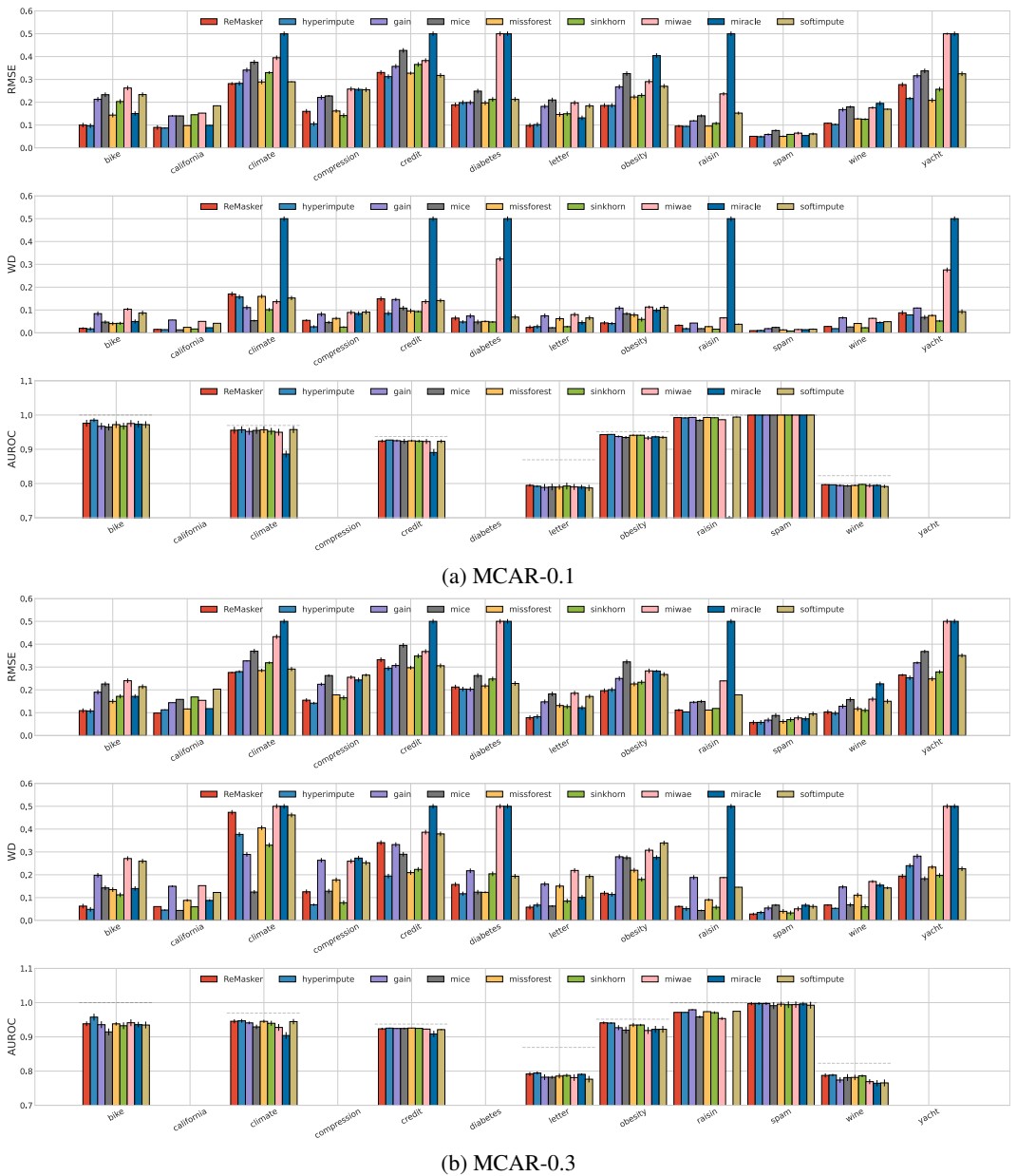

(a) MCAR-0.1

(b) MCAR-0.3

Figure 7a: Overall performance of REMASKER and 8 baselines on 12 benchmark datasets under MCAR scenario with 0.1 and 0.3 missingness ratio. The results are shown as the mean and standard deviation of RMSE, WD, and AUROC scores (AUROC is only applicable to datasets with classification tasks).

Table 7. Ablation study of REMASKER on the `california` dataset. The default setting is as follows: encoder depth = 6, decoder depth = 4, embedding width = 32, masking ratio = 50%, and training epochs = 600.

| depth | RMSE | WD |
|---|---|---|
| 2 | 0.0783 | 0.0230 |
| 4 | 0.0663 | 0.0172 |
| 6 | 0.0821 | 0.0225 |
| 8 | 0.0834 | 0.0244 |
| 10 | 0.0726 | 0.0196 |

(a) Decoder depth

| width | RMSE | WD |
|---|---|---|
| 16 | 0.0678 | 0.0213 |
| 32 | 0.0663 | 0.0172 |
| 64 | 0.0974 | 0.0322 |
| 128 | 0.1125 | 0.0388 |
| 256 | 0.0877 | 0.0324 |

(b) Embedding width

| depth | RMSE | WD |
|---|---|---|
| 2 | 0.0886 | 0.0334 |
| 4 | 0.0738 | 0.0203 |
| 6 | 0.0663 | 0.0172 |
| 8 | 0.0878 | 0.0322 |
| 10 | 0.0776 | 0.0239 |

(c) Encoder depth

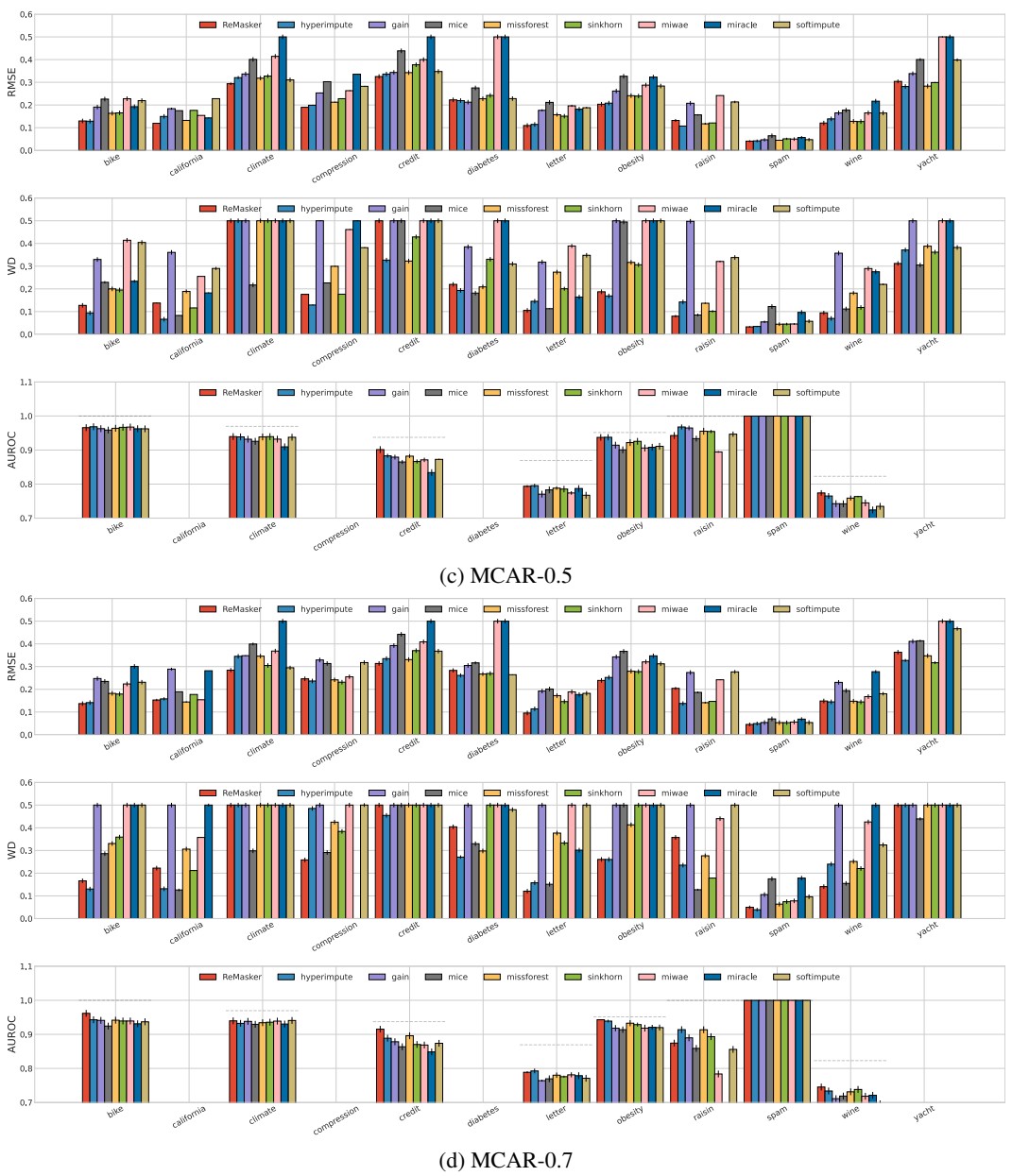

(c) MCAR-0.5

(d) MCAR-0.7

Figure 7b: Overall performance of REMASKER and 8 baselines on 12 benchmark datasets under MCAR scenario with 0.5 and 0.7 missingness ratio. The results are shown as the mean and standard deviation of RMSE, WD, and AUROC scores (AUROC is only applicable to datasets with classification tasks).

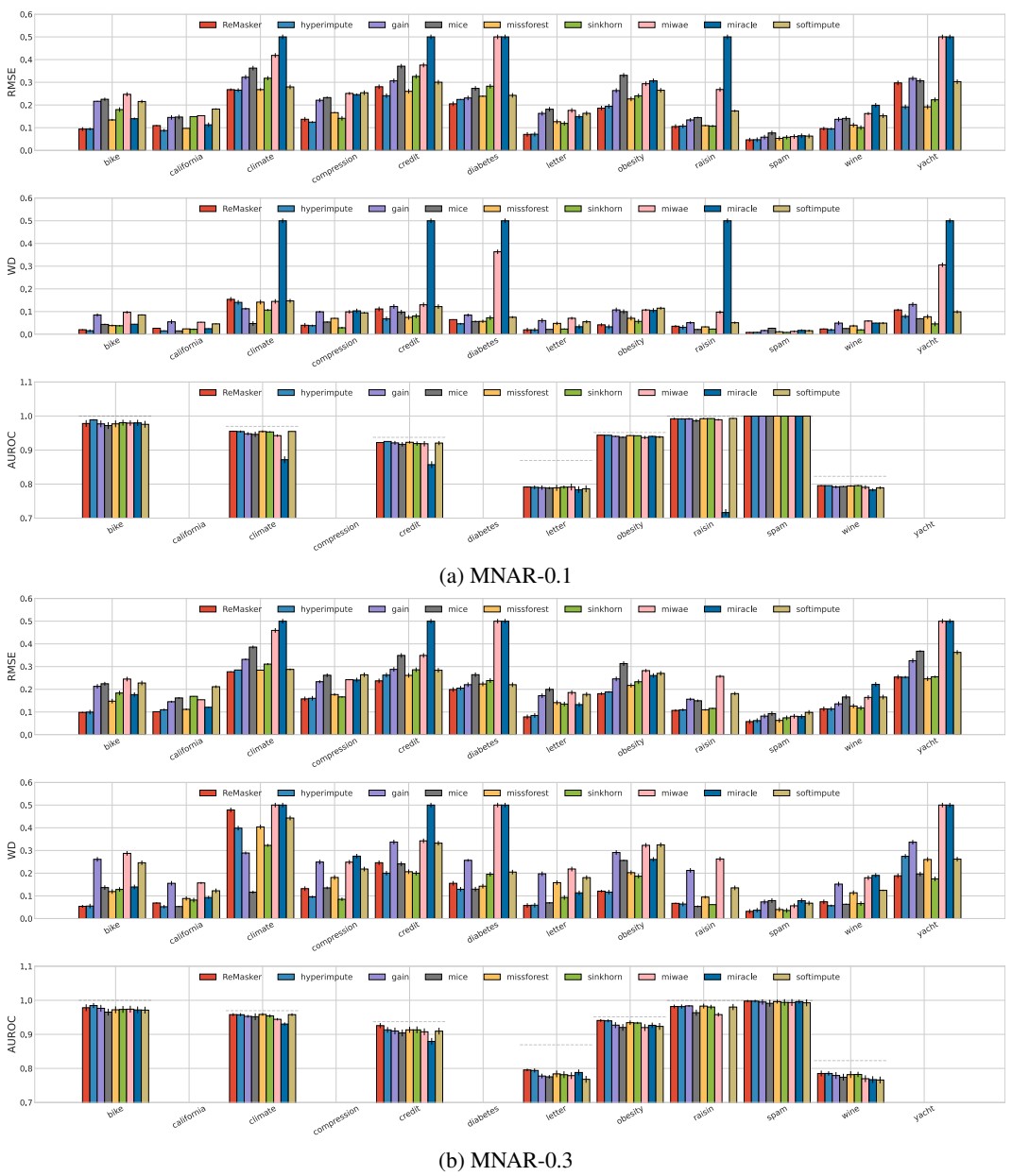

(a) MNAR-0.1

(b) MNAR-0.3

Figure 8a: Overall performance of REMASKER and 8 baselines on 12 benchmark datasets under MNAR with 0.1 and 0.3 missingness ratio. The results are shown as the mean and standard deviation of RMSE, WD, and AUROC scores (AUROC is only applicable to datasets with classification tasks).

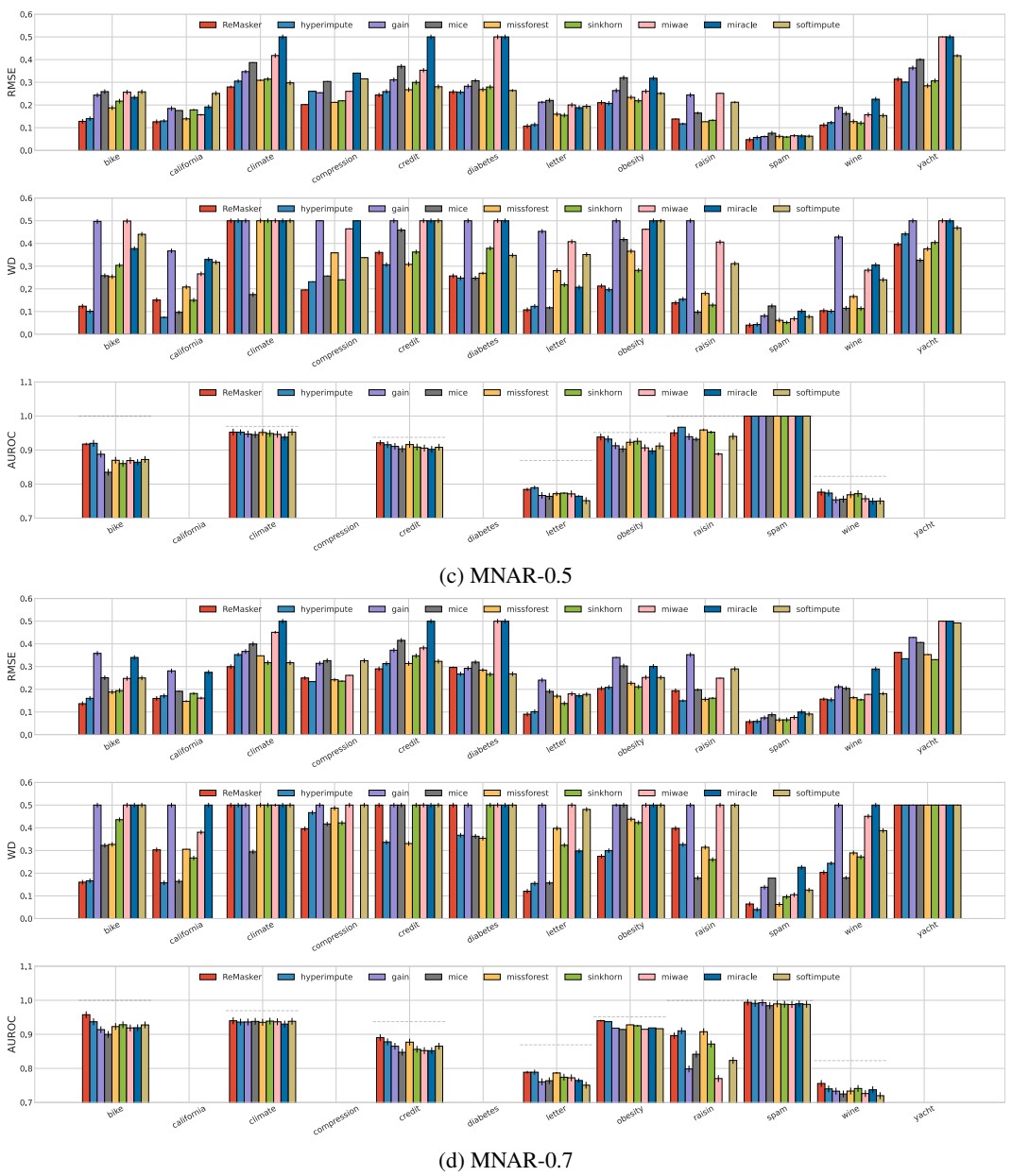

(c) MNAR-0.5

(d) MNAR-0.7

Figure 8b: Overall performance of REMASKER and 8 baselines on 12 benchmark datasets under MNAR with 0.5 and 0.7 missingness ratio. The results are shown as the mean and standard deviation of RMSE, WD, and AUROC scores (AUROC is only applicable to datasets with classification tasks).

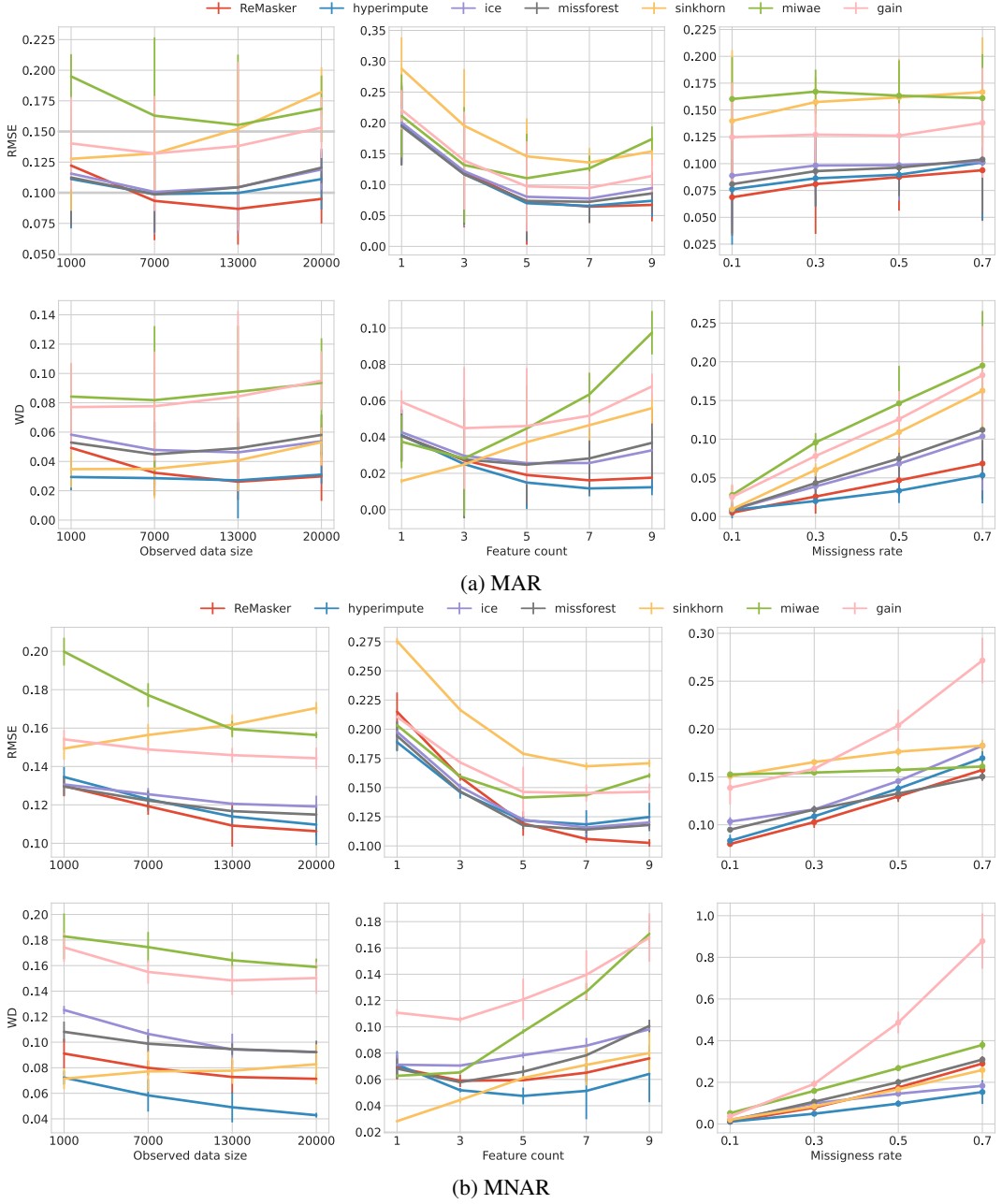

Figure 9a: Sensitivity analysis of REMASKER on the `california` dataset under MAR and MNAR scenarios. The results are shown in terms of RMSE and WD, with the scores measured with respect to (a) the dataset size, (b) the number of features, and (c) the missingness ratio. The default setting is as follows: dataset size = 20,000, number of features = 9, and missingness ratio = 0.3.

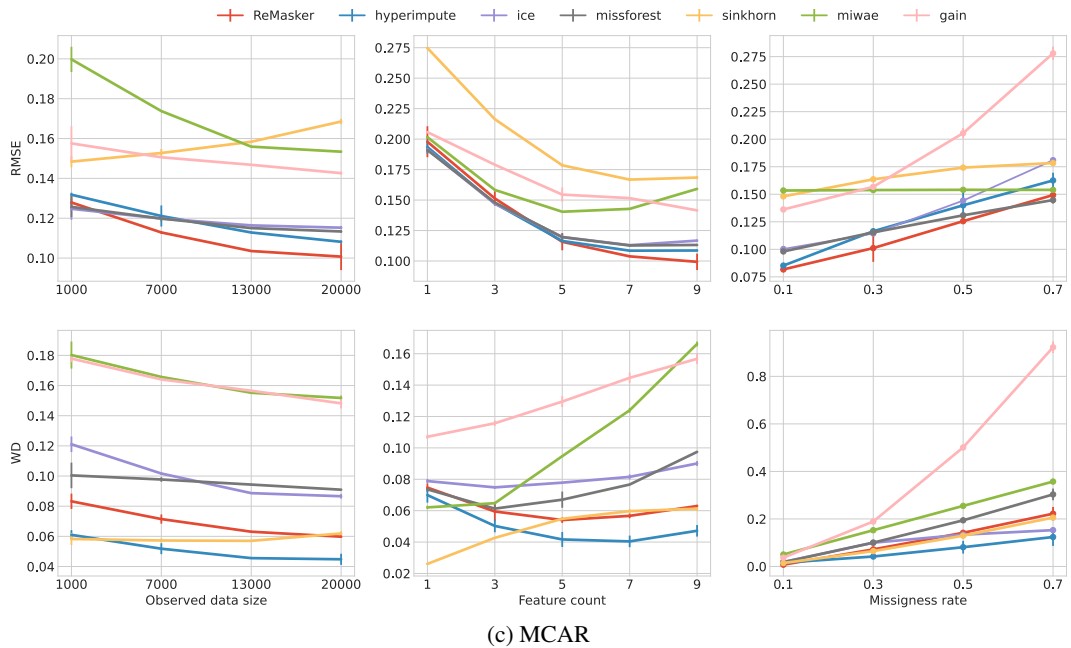

(c) MCAR

Figure 9b: Sensitivity analysis of REMASKER on the `california` dataset under the MCAR setting. The results are shown in terms of RMSE and WD, with the scores measured with respect to (a) the dataset size, (b) the number of features, and (c) the missingness ratio. The default setting is as follows: dataset size = 20,000, number of features = 9, and missingness ratio = 0.3.

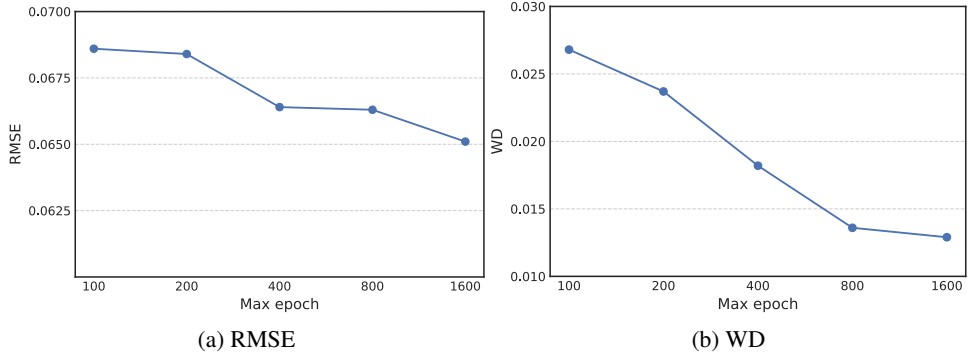

(a) RMSE                                        (b) WD

Figure 10: REMASKER performance with respect to the number of training epochs on the `california` dataset under MAR with 0.3 missingness ratio.

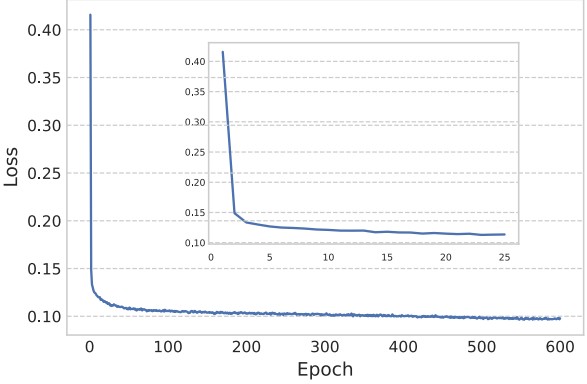

Figure 11: Convergence of REMASKER's fitting on `california` under MAR with 0.3 missingness ratio.

