# OpenReview forum: "Have Missing Data? Make It Miss More! Imputing Tabular Data with Masked Autoencoding"
_ICLR.cc/2023/Conference — Submitted to ICLR 2023_

### Official Review · Reviewer_Hron · 2022-10-19

**Confidence:** 4
**Correctness:** 3
**Technical Novelty And Significance:** 3
**Empirical Novelty And Significance:** 3
**Recommendation:** 5

**Clarity, Quality, Novelty And Reproducibility:**

The paper is nicely written and the idea is clearly explained. My only suggestion is to improve the description of Transformer model applied in ReMasker. The description provided on page 4 would be insufficient for me to re-implement the proposed approach. I suggest the authors make clear how the Transformed is implemented in this case because this is the most important part of the paper for me.

**Strength And Weaknesses:**

The idea of the proposed method is very simple, which is its great advantage. However, I have a feeling that similar ideas (masking out an additional set of values and optimizing the loss for their reconstruction) appeared in previous works, even if the authors did not sell it as a "new imputation method". When missing values do not appear in training, then an analogical approach was implemented in ContextEncoder (https://arxiv.org/abs/1604.07379). Since the authors deal with inpainting, the training data does not contain missing values. More importantly, re-masking approach was used as an ingredient in MisConv approach, see page 12 (paragraph "Training of DMFA on incomplete data") in https://arxiv.org/pdf/2110.14010.pdf. Instead of reconstruction loss, they apply negative log-likelihood since they deal with probability distributions. I think that there are more techniques with the analogical idea.

Nevertheless, I see some important contributions in the paper. This is the first paper where I see the application of the Transformer model in the case of missing data imputation. Even if the general strategy of re-masking was known, the authors show that the application of Transformed allows for obtaining significant improvement over SOTA. The authors could focus more on this aspect than on promoting a re-masking strategy.

I appreciate the very good evaluation presented in the paper, which includes three missing scenarios, and various missing ratios. Moreover, the authors compared their method with a large number of recent imputation techniques showing very good results of ReMasker.

I wonder if the authors verified what happens if we replace the Transformer model with e.g. fully connected neural network. In other words, I would like to know how much we gain by using the general strategy of re-masking with an arbitrary model and how much of the improvement is gained using the Transformer model itself.

**Summary Of The Paper:**

The paper proposes the imputation strategy called ReMasker for filling in missing values in tabular data. The idea is based on masking out randomly selected values and training the neural network to reconstruct them. As a neural network, the authors use Transformer, which better reflects the correlation between attributes. The evaluation was performed on typical tabular datasets and compared with SOTA imputation techniques. Additionally, the authors justify that REMASKER learns representations of tabular data, which are insensitive to missing values.

**Summary Of The Review:**

To summarize, I like the paper and I think it is a valuable contribution. In particular, the application of the Transformer is a major contribution for me. The authors were able to improve the SOTA in the imputation task. However, I have a feeling the idea of re-masking appeared before. In consequence, this is a new instantiation of the previous idea. This is a borderline paper for me and I am willing to increase my score after the rebuttal.

---

> ### Author Response · Authors · 2022-11-18
> **Response to Reviewer Hron**
>
> Thank you for the valuable feedback on improving this paper! Below please find our answers to your questions. More details can be found in the revised paper.
>
> > However, I have a feeling that similar ideas (masking out an additional set of values and optimizing the loss for their reconstruction) appeared in previous works, even if the authors did not sell it as a "new imputation method".
>
> Thank you for the insightful comments. While the idea of "re-masking" is not new in general, the novelty of this work lies in three major aspects: 1) it applies the re-masking idea within the framework of masked auto-encoding; 2) it applies masked auto-encoding to model tabular data; 3) it applies Transformer in the task of tabular data imputation. We have included the missing references [1,2] in Section 2 to better contextualize our work.
>
> > I wonder if the authors verified what happens if we replace the Transformer model with e.g. fully connected neural network. In other words, I would like to know how much we gain by using the general strategy of re-masking with an arbitrary model and how much of the improvement is gained using the Transformer model itself.
>
> Thank you for the suggestion. We have included the experiments using other backbone models (details in Section 4.2). We compare the performance of ReMasker with different backbone models (i.e., Transformer, linear, and convolutional) with the number of layers and the size of each layer fixed as the default setting. As shown in Table 2, Transformer-based ReMasker largely outperforms the other variants, which may be explained by that the self-attention mechanism of Transformer is able to effectively capture the intricate inter-feature correlation.
>
> **References:**
>
> [1] ContextEncoders: Feature Learning by Inpainting
>
> [2] MisConv: Convolutional Neural Networks for Missing Data
>
> Please let us know if you have further questions/suggestions.
>
> Thanks,
>
> Authors of Paper 3141

---

### Official Review · Reviewer_CAJ5 · 2022-10-19

**Confidence:** 5
**Correctness:** 3
**Technical Novelty And Significance:** 2
**Empirical Novelty And Significance:** 2
**Recommendation:** 5

**Clarity, Quality, Novelty And Reproducibility:**

- The idea is clear and the paper is easy to understand.
- It is hard to say that the idea is novel. Mask autoencoding is widely utilized in many areas and using mask autoencoders for imputation is quite straightforward.
- It seems like the paper can be somewhat easily reproducible.

**Details Of Ethics Concerns:**

Not applicable.

**Strength And Weaknesses:**

Strength:
- The methodology is simple but the performance is quite impressive.
- The authors provided extensive experiments to empirically verified the claims.
- The authors utilize the transformer algorithm for tabular data imputation which seems like a proper model.

Weakness:
- It is unclear how the MCAR based remasking can be generalized to impute missing components with MAR and MNAR settings.
- The performances are comparable with SOTA and not consistently better than SOTA. But it seems like the authors claim that it consistently outperforms SOTA.

**Summary Of The Paper:**

- The authors proposed a simple but effective imputation framework for tabular data.
- The proposed framework additionally introduces the missingness to the incomplete data and using the reconstruction loss on the introduced missing components and unmasked components to train.
- The proposed method achieves comparable results in comparison to SOTA (HyperImpute).

**Summary Of The Review:**

1. Complete data availability
- Many of the imputation methods do not need complete data for missing data imputation.
- Therefore, it would be better to tone down this contribution.

2. Re-masking with MCAR setting
- Authors introduce the re-masking with MCAR setting.
- In other words, m' is uniformly sampled.
- In that case, it is unclear how this kind of remasking and reconstructing those remasked components can be generalized to the MAR or MNAR settings.
- More specifically, the model is learning to estimate the missing components with MCAR settings via remasking and reconstruction. How this learning process can be generalized to impute the missing values in MAR and MNAR settings.

3. SOTA performances?
- It seems like the performance of the proposed method is comparable with HyperImpute.
- However, it is hard to say which one is better.
- In that point of view, it is hard to support the claims in the abstract that "ReMasker consistently "outperforms" SOTA".

4. Categorical data
- In tabular data learning, handling categorical data is critical.
- In this work, how to impute the categorical data?
- What kind of embedding did the authors use for handling categorical data?

5. Hyper-parameters
- It seems like some of the hyper-parameters are important.
- For instance, the missing ratio of remask would be a critical hyper-parameter.
- Also, the reconstruction loss weights between remask and unmask can be another important hyper-parameter.
- It would be good to add additional sensitivity analyses on those hyper-parameters.

---

> ### Author Response · Authors · 2022-11-18
> **Response to Reviewer CAJ5**
>
> Thank you for the valuable feedback! Below please find our answers to your questions. More details can be found in the revised paper.
>
> > Many of the imputation methods do not need complete data for missing data imputation. Therefore, it would be better to tone down this contribution.
>
> Thank you for the suggestion. We have revised the text as "Some of these methods either require complete data during training or operate on the assumptions of specific missingness patterns."
>
> > More specifically, the model is learning to estimate the missing components with MCAR settings via remasking and reconstruction. How this learning process can be generalized to impute the missing values in MAR and MNAR settings.
>
> Thank you for this insightful question. ReMasker is not designed specifically for the MCAR setting. Regardless of the missingness mechanism, it is rare that the values of one feature $x$ are missing across all the records. Thus, by its design, ReMasker is able to learn to re-construct feature $x_i$ conditional on other features $x_{\bar{i}} = (x_1, \ldots, x_{i-1}, x_{i+1}, \ldots, x_d)$. Yet, as reflected in the imputation results, the learning to re-construct performs better under MCAR, in which the missing values are evenly distributed across different features, than MAR or MNAR, in which the missing values are not evenly distributed.
>
> > In that point of view, it is hard to support the claims in the abstract that "ReMasker consistently "outperforms" SOTA".
>
> We have revised the text as "we show that ReMasker performs on par with or outperforms state-of-the-art methods in terms of both imputation fidelity and utility under various missingness settings".
>
> > In tabular data learning, handling categorical data is critical. In this work, how to impute the categorical data? What kind of embedding did the authors use for handling categorical data?
>
> Thank you for the question. In the current implication, we use one-hot encoding as the initial embedding for categorical attributes.
>
> > It seems like some of the hyper-parameters are important. For instance, the missing ratio of remask would be a critical hyper-parameter. Also, the reconstruction loss weights between remask and unmask can be another important hyper-parameter. It would be good to add additional sensitivity analyses on those hyper-parameters.
>
> We have conducted the sensitivity analysis of both hyper-parameters in Sections 4.2 and 4.3. We measure the performance of ReMasker under three different settings of the construction loss: (i) $\mathcal{I}_\mathrm{mask+} \cup \mathcal{I}_\mathrm{unmask} $, (ii)  $\mathcal{I}_\mathrm{mask+}$ only, and (iii) $\mathcal{I}_\mathrm{unmask}$ only on the letter and california datasets, with results shown in Table 3. It is observed that using the reconstruction of unmasked values only is insufficient and yet including the reconstruction loss of unmasked values improves the performance. We measure the impact of the masking ratio on the performance of ReMasker, with results shown in Table 4a. It is observed that the optimal ratio differs across different datasets, which may be explained by the varying number of features of different datasets (16 versus 9 in letter and california). Intuitively, a larger number of features affords a higher masking ratio to balance (i) encouraging the model to learn missingness-invariant representations and (ii) having sufficient supervisory signals to facilitate the training.
>
> Please let us know if you have further questions/suggestions.
>
> Thanks,
>
> Authors of Paper 3141

---

> > ### Comment · Reviewer_CAJ5 · 2022-12-06
> > **Thank you for the response**
> >
> > Thank you for the authors' careful responses.
> >
> > Some comments are well addressed in the rebuttals and revised manuscript.
> > - Other imputation methods without complete data assumption,
> > - Categorical features
> > - Additional sensitivity analyses
> >
> > But still, there are some remaining concerns in this paper.
> > (1) Introduce missingness with MCAR setting
> > - As I explained in the previous comment, the authors "introduce" additional missingness via MCAR. Then, the model reconstructs those additional missing components. In that case, intuitively, it is unclear whether this can be generalized to MAR and MNAR.
> > (2) Similar performance with SOTA
> > - In that case, the value of this paper is reduced.
> >
> > Therefore, it is hard for me to increase my score. Thus, I am standing to my original score (5).
> >
> > Thank you again for carefully addressing my comments.

---

### Official Review · Reviewer_Vv4x · 2022-10-23

**Confidence:** 4
**Correctness:** 2
**Technical Novelty And Significance:** 1
**Empirical Novelty And Significance:** 2
**Recommendation:** 3

**Clarity, Quality, Novelty And Reproducibility:**

The paper is clear. Figure 1 is nice-looking and clearly explains the method. The novelty is limited.

**Strength And Weaknesses:**

Strength:
1. The authors use many datasets and include many baselines (12 datasets and make a comparison with 13 imputation methods).
2. The authors does ablation study and has some discussion trying to explain why their method works.

Weakness:
1. The method does not seem to be novel. This is the main weakness. The "remask" method basically masks out some extra non-missing features and predict them. But this kind of technique is widely used in imputation method in real-world dataset when there is missingness. Researchers usually mask out extra non-missing features to create datasets for training when there is already missingness in real-world data. So I am not sure what extra novelty the paper provides.
2. The authors only consider synthetic dataset where the authors create missingness from some pre-specified missingness patterns on data where. The authors should consider real-world dataset where there is already missingess.
3. The discussion on learning representation invariant to missingness pattern is vague. All the methods try to learn representation invariant to missingness patterns. The authors' analysis does not provide extra intuition.


**Summary Of The Paper:**

The paper proposes a remask method which masks out some more data under the original missing data and reconstruct them. The main contribution is this remask part.

**Summary Of The Review:**

The main reason I choose to reject is that the novelty of the paper is limited.

---

> ### Author Response · Authors · 2022-11-18
> **Response to Reviewer Vv4x**
>
> Thank you for the valuable feedback! Below please find our answers to your questions. More details can be found in the revised paper.
>
> > The method does not seem to be novel. This is the main weakness. The "remask" method basically masks out some extra non-missing features and predict them. But this kind of technique is widely used in imputation method in real-world dataset when there is missingness. Researchers usually mask out extra non-missing features to create datasets for training when there is already missingness in real-world data. So I am not sure what extra novelty the paper provides.
>
> Thank you for the comments. While the idea of "re-masking" is not new in general, the novelty of this work lies in three major aspects: 1) it represents the first work of applying the re-masking idea within the framework of masked auto-encoding; 2) it applies masked auto-encoding to model tabular data; 3) it applies Transformer in the task of tabular data imputation. Overall, our findings indicate that, besides its success in the language and vision domains, masked modeling also represents a promising direction for future research on tabular data imputation.
>
> > The authors only consider synthetic dataset where the authors create missingness from some pre-specified missingness patterns on data where. The authors should consider real-world dataset where there is already missingess.
>
> To measure the performance of an imputation method, it is essential to have the ground-truth values of missing data. Due to their lack of ground-truth values, it is not suitable to use real-world datasets with missing values directly. Thus, following prior work, we use benchmark datasets with synthesized missingness in our evaluation.
>
> > The discussion on learning representation invariant to missingness pattern is vague. All the methods try to learn representation invariant to missingness patterns. The authors' analysis does not provide extra intuition.
>
> Thank you for the comments. Although many imputation methods attempt to learn missingness-invariant latent representations of tabular data, few have analytically and empirically analyzed this property. To our best knowledge, this work is the first to explore this property in the context of masked auto-encoding-based tabular data imputation.
>
> Please let us know if you have further questions/suggestions.
>
> Thanks,
>
> Authors of Paper 3141

---

### Official Review · Reviewer_socG · 2022-10-25

**Confidence:** 2
**Correctness:** 3
**Technical Novelty And Significance:** 3
**Empirical Novelty And Significance:** 3
**Recommendation:** 6

**Clarity, Quality, Novelty And Reproducibility:**

Clarity
--------
-Aside from a few points (see weaknesses), the paper is well written and easy to follow.

Quality
--------
-The paper contains significant contributions for ML applications suffering from missing data.

Novelty
---------
-Leveraging MAE, ReMasker is a novel contribution for tabular data imputation.  The inclusion of a transformer is also novel, but an ablation study or discussion of why this (and inclusion of the decoder during training) is currently lacking in the paper.

Reproducibility
-----------------
Results are not reproducible, no code was included.

**Strength And Weaknesses:**

Strengths:
-ReMasker is a great extension of MAE to tabular data imputation

-ReMasker works well across different downstream performance metrics (i.e., RMSE for regression and AUAOC for classification), outperforming many widely used imputation schemes (e.g., MissForest and MICE).

-The ReMasker architecture is pretty lightweight, making it a reasonable approach to large-scale tabular datasets

Weaknesses:
-Important model design choices are not explained in any depth.  E.g., "Unlike conventional MAE, the REMASKER decoder is used in both fitting and imputation phases." <- Why?  Without explanations for the differences in models between MAE and ReMasker, an evaluation must be conducted to explain these decisions.

-The exact model details could be better explained.  E.g., "Note that the encoder is only applied to the observed values: in the
 fitting phase, it operates on the observed values after re-masking" <- This statement is inconsistent with the paper; based on Figure 1,
 it is operating on missing and masked values, which are not observed.  Please clarify

-More work is needed to show that ReMasker theoretically learns missing-invariant representation of tabular data (in Section 5).  The current section is unconvincing

-Only MAR is considered in the main text.  A discussion of the MCAR and MNAR results included in the appendix would greatly strengthen the paper.

-The ablation study is limited; it is currently an evaluation of network-architecture hyperparameters.  An actual ablation study would consider the role of transformers in ReMasker (vs without in the original MAE architecture) and the exclusion of the decoder during training (as is originally done in the MAE for vision).

**Summary Of The Paper:**

The authors propose ReMasker, an adaptation of the masked autoencoder (MAE, for vision tasks) for tabular data imputation.  Additionally, ReMasker included transformer blocks to the autoencoder and both the encoder+decoder during training.  Missing at random (MAR) is simulated for a large number (12) of datasets, and the utility of imputation using ReMasker is evaluated for different metrics, as well as in contrast to a large number of SOTA imputation methods.  ReMasker regularly outperforms the other baseline methods.  Additional studies include the effects of the network architecture (i.e., depth and width) on overall performance, the masking ratio hyperparameter, and the combination of ReMasker with the ensemble imputer HyperImpute.

**Summary Of The Review:**

Results are impressive and the use of MAE for tabular data imputation is intuitive.  However, the results are not currently reproducible and the paper could clarify more points (detailed throughout the review).  I currently lean towards accept.

---

> ### Author Response · Authors · 2022-11-18
> **Response to Reviewer socG**
>
> Thank you for the valuable feedback! Below please find our answers to your questions. More details can be found in the revised paper.
>
> > Results are not reproducible, no code was included.
>
> We have included the code of this work at the following anonymous GitHub link: https://anonymous.4open.science/r/remasker-5E3B
>
> > Important model design choices are not explained in any depth. E.g., "Unlike conventional MAE, the REMASKER decoder is used in both fitting and imputation phases."
>
> Conventional MAE focuses on representation learning and uses the decoder only in the training phase. In ReMasker, the decoder is required to re-construct the missing values and is thus used in both fitting and imputation phases.
>
> > The exact model details could be better explained. E.g., "Note that the encoder is only applied to the observed values: in the fitting phase, it operates on the observed values after re-masking" <- This statement is inconsistent with the paper; based on Figure 1, it is operating on missing and masked values, which are not observed. Please clarify.
>
> Thank you for the question. We have revised Figure 1 to clarify that in the fitting phase, the encoder operates on the observed values after re-masking; in the imputation phase, it operates on all the observed values.
>
> > Only MAR is considered in the main text. A discussion of the MCAR and MNAR results included in the appendix would greatly strengthen the paper.
>
> Thank you for the suggestion. We have included a discussion about the MCAR and MNAR results in Appendix B.1. More concretely, ReMasker performs well under all the missingness mechanisms, among which, it performs the best under MCAR. We have the following explanations. Regardless of the missingness mechanism, it is rare that the values of one feature $x$ are missing across all the records. Thus, by its design, ReMasker is able to learn to re-construct feature $x_i$ conditional on other features $x_{\bar{i}} = (x_1, \ldots, x_{i-1}, x_{i+1}, \ldots, x_d)$. Yet, as reflected in the imputation results, the learning to re-construct performs better under MCAR, in which the missing values are evenly distributed across different features, than MAR or MNAR, in which the missing values are not evenly distributed.
>
> > The ablation study is limited; it is currently an evaluation of network-architecture hyperparameters. An actual ablation study would consider the role of transformers in ReMasker (vs without in the original MAE architecture) and the exclusion of the decoder during training (as is originally done in the MAE for vision).
>
> Thank you for the suggestion. We have included the experiments using other backbone models (details in Section 4.2). We compare the performance of ReMasker with different backbone models (i.e., Transformer, linear, and convolutional) with the number of layers and the size of each layer fixed as the default setting. As shown in Table 2, the Transformer-based ReMasker largely outperforms the other variants, which may be explained by that the self-attention mechanism of Transformer is able to effectively capture the intricate inter-feature correlation under limited data.
>
> Please let us know if you have further questions/suggestions.
>
> Thanks,
>
> Authors of Paper 3141

---

### Decision · Program_Chairs · 2023-01-20

**Decision:**

Reject

**Justification For Why Not Higher Score:**

The contribution is essentially empirical, but not rigorously executed.

**Justification For Why Not Lower Score:**

N/A

**Metareview: Summary, Strengths And Weaknesses:**

The paper deal with learning on tabular data with missing values and proposes ReMasker, a variant of masked autoencoders where the model is trained to reconstruct missing entries that have been masked completely at random during training. In addition to this architectural variant the authors perform a series of experiments and ablation tests on a set of UCI datasets under the MAR assumption.

During reviews, a number of concerns were raised, one of the most prominent ones (shared across reviewers) being the lack of novelty for the (re)masking operation in autoencoders and Transformers to deal with missing values. During the discussion, it emerged that the contribution can be found in applying this remasking technique specifically to tabular data. Nevertheless, the maior issue behind applying this heuristic is the lack of theory behind the interactions between MCAR remasking and the actual missingnes mechanism in the data. While missingness mechanisms have been investigated in breadth and depth in these years, heuristics of this kind can still be very useful in practice. However, the empirical gains of the proposed ReMasker does not seem to make a clear case over the 12 datasets used and under the synthetic MAR assumption.

As reviewers highlighted, the paper would greatly improve if the contribution would be better placed in the burgeoning literature of missing values as well as Transformers. Furthermore, an extended experimentation on different missing mechanisms and on more challenging datasets would better help understand the effectiveness of ReMasker. Statistical tests should be used to rigorously assess the improvements over alternative missing value imputation schemes.

The paper is rejected.

**Summary Of Ac-Reviewer Meeting:**

During rebuttal, we quickly converged towards rejection and there was no meeting.